# Death-coping self-efficacy and its influencing factors among Chinese nurses: A cross-sectional study

Xi Lin[1,2,3], Xiaoqin Li[1,2], Yongqi Bai[1,2]*, Qin Liu[1,2,3]*, Weilan Xiang[4]

**1** Department of Pediatrics, The Affiliated Hospital of Southwest Medical University, Luzhou, Sichuan, China, **2** Sichuan Clinical Research Center for Birth Defects, Luzhou, Sichuan, China, **3** Department of Nursing, The Affiliated Hospital of Southwest Medical University, Luzhou, Sichuan, China, **4** Department of Nursing, Zhejiang University School of Medicine Sir Run Run Shaw Hospital, Hangzhou, Zhejiang, China

* luzhou1991@yeah.net (YB); liuqin18@126.com (QL)

**Data Availability Statement:** All relevant data are within the paper and its Supporting Information files.

**Funding:** The authors received no specific funding for this work.

## Abstract

### Background

Nurses are the main caregivers of dying patients. Facing or dealing with death-related events is inevitable. Death-coping self-efficacy (DCS) is very important, as it can reduce the risk of nursing staff to adverse emotional distress, help them participate in end-of-life care and improve the quality of care of patients.

### Methods

Using the convenient sampling method, this study included a total of 572 nurses from a tertiary hospital in Hangzhou, China. The status and influencing factors of the DCS of nurses were explored using a general information questionnaire and DCS scale.

### Results

The scores of each parameter, ranging from low to high, were in the order of coping with grief, preparation for death and hospice care. Factors influencing nurses' DCS included attendance in hospice care education courses within the previous year, experience of accompanying the family members of the deceased and attitude towards death.

### Conclusions

The overall self-efficacy of nurses in palliative care was at a medium level. Moreover, their self-efficacy in coping with grief and preparation for death should be strengthened. Managers of medical institutions can assess the death-coping ability of nurses, which helps provide corresponding support and training for nurses at an early stage. Nurses should receive guidance in grief adjustment and emotion regulation. Medical units should provide nurses with a platform for continuous training and education, use of death-related theoretical models and frameworks to guide nurses in dealing with death-related events, reduce nurses' negative mood and jointly promote their mental health.

**Competing interests:** NO authors have competing interests

**Abbreviations:** DCS, Death-coping self-efficacy; DCSS, Death-coping self-efficacy scale.

# Introduction

Death-coping self-efficacy (DCS) refers to whether a nurse has the confidence to provide palliative care timely to dying patients and assist their family members, which include the confidence to evaluate needs, dealing with care problems, managing symptoms and providing information, coping with the death of a loved one and planning and preparation for burial [1]. It has always been the core goal of palliative care to allow patients to treat death with the best mentality, optimism and happiness at the end of life, and to achieve a "good death" [2]. When caring for terminally ill patients and dealing with death-related events, it often brings different degrees of anxiety, fear, fatigue, helplessness, guilt and other negative moods to nurses, which affect nurses' daily life and even work quality [2, 3]. Managers of medical institutions and patients generally believe that dealing with death-related events is a special mission that medical staff should undertake, and often ignore the psychological stress and trauma brought to medical staff, especially front-line nurses, in the process. DCS can assist nurses to better participate in the development of palliative care strategies, the arrangement of end-of-life care programs, death preparation and planning, and death dialogue, and are essential skills for nursing professionals. Whether nursing students are on clinical traineeships, internships, or entering clinical positions after graduation, if nurses have insufficient understanding and preparation for death, they may increase the risk of negative attitudes such as burnout. Nurses' frequent exposure to patient death-related events can also lead to compassion fatigue and burnout [4, 5]. Even among the medical personnel who know about palliative care, they still equate palliative care with 'giving up' or shortening of their life span [6] and saving the lives of patients with terminal illness 'at all costs' [7]. This indicates that China lacks professional palliative care personnel who can provide palliative care services.

Zheng et al. [2] investigated the palliative care behaviour and self-efficacy of 338 nurses and found that the number of patients cared for did not affect the self-efficacy of nurses; however, the willingness and emotions of patients directly affected the self-efficacy and work execution of nurses. Yang et al. [8], scholars from Taiwan, investigated the relationship between nurses' emotional distress and self-efficacy in coping with death while taking care of dying patients. The results showed that young and junior nurses exhibited higher emotional distress, which negatively correlated with their self-efficacy in coping with death. A previous study reported that knowledge and nursing experience in palliative care are key factors in improving self-efficacy [9]. The literature has also pointed out that most of the nurses are aware of the insufficient education and training in death-related and end-of-life care. Specifically, the new staff in the intensive care unit, who take care of patients with complex diseases, often feel that they are unable to care for patients at the end of their lives, resulting in physical and mental exhaustion, poor working ability and increased staff turnover rate [10, 11]. The emergency medical staff generally considers that patients are exposed to the emergency unit for a short period; therefore, there are practical difficulties in providing end-stage care to them [11]. The main obstacles for nurses in caring for patients at the end of their lives include their low awareness of palliative care guidance, lack of education and training for palliative care by hospitals and most of the nurses do not participate in palliative care education courses, thereby affecting their nursing guidance ability [12]. According to Zheng et al. [13], working years, behavioural experience of end-stage nursing and perceived importance of nurses are positively correlated with the self-efficacy of palliative care. However, Pfister et al. [14] found that the palliative care knowledge and self-efficacy were positively and negatively correlated with work experience, respectively. Other studies have shown that nurses with relevant work experience, aged >36 years, were female and had formal training in palliative care scored higher in communication self-efficacy evaluation [15]. Based on the above literature, age, sex, work experience, work

unit, willingness and attitude of nurses to care for patients, knowledge and attitude to palliative care and other factors might affect their self-efficacy, but the correlation between some variables remains different.

Improving the self-efficacy of nurses to cope with the death of patients is important. Lien et al. [16] concluded that dying patients can feel care and satisfaction from the patience, care and comfort of nurses. Hospital clinical nurses' nursing ability and attitude to care for terminally-ill patients directly affect the nursing quality of terminally-ill patients [17]. However, studies have shown that Chinese nurses have low levels of attitudes and cognitions towards death [2]. At present, most studies on death focus on the attitude level of nurses, while a few empirical studies have evaluated the self-efficacy of nurses. Therefore, this study intended to analyse the status quo of the self-efficacy of nurses to cope with patient's death and discussed its influencing factors. The findings of this study will serve as a reference for human resource management, utilisation of medical institutions and search for the stability of nurses' manpower and will contribute to the improvement of the quality of nurses' work and palliative care.

## Methods

### Study design and participants

The Epidemiological Observational Study Report guidelines and methodologies were used in the reporting of the results of this study.

A cross-sectional survey was conducted using a questionnaire survey among registered nurses in a tertiary hospital in Hangzhou, China, from August to September of 2020 through convenience sampling. First, participants were briefed on the purpose of the study, and their verbal consent was obtained. Notably, nurses have cared for patients with terminal illness. The authors obtained information that could identify individual participants after data collection.

Inclusion criteria: ①Nurses who have cared for terminally ill patients; ②Informed consent and willing to cooperate with the investigator; ③Clinical nurses who hold a nurse qualification certificate and are registered on the job.

Exclusion criteria: ①Clinical nurses on shifts, advanced studies or going out; ②Nurses on sick leave, maternity leave or vacation

The G-Power 3.1 statistical software was used to estimate the sample size, with two-tailed tests, effect size of 0.3, power of 0.95 and α value of 0.05, which was calculated to be 134. Considering the possibility for dropped or missed out cases during the study, an additional 10% of the study participants were selected, and the total number of participants selected as a sample was 250. To reduce the sampling error and make the conclusions more reliable, the sample size was set to 600. After excluding 28 incomplete responses, a total of 572 questionnaires were included in the analysis.

### Instruments

**General information questionnaire.**   A total of 10 parameters were investigated, which included sex, age, educational background, department, marital status, religious belief, duration of clinical work, attendance in palliative care education courses within the previous year, experience of accompanying the family members of the deceased and attitude towards death. Participation in the palliative care education course was defined as taking part in online or offline training on palliative care-related content, in which nurses participated in each training session for not less than 40 min.

**Death-coping self-efficacy scale (DCSS).**   The DCSS was originally developed by Robbins [18], an American scholar, for hospice wards in 1992. It had good reliability and validity. In

2006, Professor Zhang [19], a Taiwanese scholar, compiled the DCSS in hospice wards and formed its Taiwanese version, which has been used widely as a measurement tool to study DCS in Taiwanese nurses [5].

The DCSS has 29 questions and three dimensions, including 12 questions on hospice care. It evaluates nurses' confidence in providing physical and spiritual care of dying patients and their families. Nine questions are on grief management, which assesses nurses' confidence in dealing with their grief when facing the death of other people. There were eight questions on the preparation for death, which assess the caregivers' confidence in planning their death preparation. In this study, the expert validity test value of DCSS was 0.97, and the scores for the three subscales ranged from 0.857 to 0.893. Cronbach's α value of the total scale was 0.905. A 5-point Likert scale was adopted, in which 1 point indicates surely not and 5 points mean surely yes; the total score was 29–145 points. The scores for the questions in each dimension were added and taken as the score of the nurses' ability to cope with patients' death. The higher the score, the more self-efficacy they had in coping with death and vice versa. In this study, the Taiwanese version of DCSS was used to evaluate nurses' DCSS score.

## Data collection

For this study and relevant literature, the questionnaire survey was adopted as the method for data collection. The investigator and two nurses conducted a field investigation in each department to distribute the questionnaires and explained the study purpose and method of questionnaire collection for the data collection. Participation is voluntary, and all responses are anonymous. In the course of the study, all relevant data and questionnaire contents were first coded and then recorded by a computer. The answer time was 8–15 min. After the questionnaires were collected, the answers were reviewed and sorted. Valid questionnaires were then coded, and data were processed using a quantitative method.

## Ethical consideration

This study was approved by the Research Ethics Committee of the Sir Run Shaw Hospital, College of Medicine, Zhejiang University (No. 20201029–31). Written informed consent was obtained from all participants in this study, the data were strictly limited, and they were also assured that these questionnaires were used for research purposes only.

## Quality control

To avoid bias in responses, data about the training purpose of the researchers, issues that needed attention and methods of questionnaire collection were determined before the questionnaires were distributed.

The integrity of the returned questionnaires was checked, and invalid ones were excluded. The exclusion criteria of the questionnaires were as follows: all or basic questions had the same answers, >10% of the answers were missing and obvious logical errors were noted in the responses. After the questionnaire was collected on the spot, the integrity and logicality of the questionnaire were checked, and the missing answers <10% were corrected in time.

## Data analysis

After the collection of questionnaires, data were imported to EpiData3.1, and the IBM SPSS Statistics 26.0 (IBM Corp., Armonk, NY, USA) was used as the main statistical analysis tool. The analysis methods were as follows: general information of the nurses and DCSS scores were described as mean, percentage and standard deviation. The Chi-square test, independent

sample *t*-test, one-way analysis of variance and multiple post-mortem comparison tests were used to analyse differences in sex, age group, titles, departments and other demographic data. The multiple linear stepwise regression analysis was used to analyse factors influencing nurses' DCS. All tests were conducted on both sides. The test level $\alpha = 0.05$ and $P<0.05$ indicated significant differences.

## Results

A total of 600 questionnaires were sent, and 594 were recovered. Incomplete and invalid questionnaires were removed, leaving 572 valid questionnaires with a valid questionnaire rate of 95.3%.

### Sociodemographic and work characteristics of the participants

Of the 572 nurses, 569 (99.5%) were women, and 57.2% were married. The mean age of the nurses was $32.4 \pm 7.1$ (range, 23–54) years. Most of the nurses (73.1%) had clinical working experience of $\leq$10 years. Moreover, 518 (90.6%) nurses had a bachelor's degree, 43 (7.5%) had a master's degree, 94.8% had no religious affiliations, 38.5% did not attend palliative care education courses within the previous year, 35.3% had experienced accompanying the family members of the deceased and 59.4% said they would accept death. Details are shown in Table 1.

### Status quo of the DCSS of nurses

Among the three dimensions of DCSS, hospice care ability recorded the highest score, while coping with grief had the lowest score. The scores of each dimension and parameter are presented in Table 2.

Table 3 presents the descriptive statistics of DCSS and its subscales, namely, hospice care, coping with grief, preparation for death and other parameters. The average DCSS score was 10.46 (*SD* 1.27), while that for the subscales were 3.95 (*SD* 0.48) for hospice care, 3.11 (*SD* 0.62) for coping with grief and 3.39 (*SD* 0.56) for preparation for death. Among the 11 parameters, 'allow a patient to communicate fully' showed the highest mean (4.30), while 'coping with the death of your child' (parameter 22) presented the lowest mean (2.41).

### Factors affecting DCSS

The DCSS scores of nurses were taken as the dependent variable, and factors showing significance in general data were taken as independent variables. Multiple stepwise regression analysis was conducted according to the levels of $\alpha = 0.05$ in the entry model and $\alpha = 0.10$ in the exit model. After the final entry into the equation, the factors were analysed for attendance in palliative care education courses within the previous year, personal bereavement experience and attitudes about death (Table 4).

### Factors affecting hospice care

The scores of nurses on the hospice care subscale were taken as the dependent variable, and factors with significance in general data were taken as independent variables. Multiple stepwise regression analysis was conducted according to the levels of $\alpha = 0.05$ in the entry model and $\alpha = 0.10$ in the exit model. After the final entry into the equation, the factors were analysed for educational background, attendance in hospice care education courses within the previous year and attitudes about death (Table 5).

**Table 1. Sociodemographic and work characteristics of the participants (N = 572).**

| Variable | Subgroup | Frequency (N) | Percentage (%) |
|---|---|---|---|
| Age | ≤30 | 348 | 60.8 |
| | 31~40 | 187 | 32.7 |
| | ≥41 | 37 | 6.5 |
| Gender | | | |
| | Male | 3 | 0.5 |
| | Female | 569 | 99.5 |
| Marital status | | | |
| | Single | 238 | 41.6 |
| | Married | 326 | 57.0 |
| | Divorced | 8 | 1.2 |
| Length of service (yrs.) | | | |
| | ≤10 | 418 | 73.1 |
| | >10 | 154 | 26.9 |
| Department | | | |
| | Surgical | 213 | 37.2 |
| | Medicine | 245 | 42.8 |
| | ICU | 28 | 4.9 |
| | Oncology ward | 52 | 9.2 |
| | Emergency | 34 | 5.9 |
| Educational background | | | |
| | Associate (College) | 11 | 1.9 |
| | Bachelors (University) | 518 | 90.6 |
| | Masters | 43 | 7.5 |
| Religious affiliation | | | |
| | Yes | 30 | 5.2 |
| | None | 542 | 94.8 |
| Attended palliative care education courses within one year | | | |
| | Yes | 352 | 61.5 |
| | No | 220 | 38.5 |
| Personal bereavement experience | | | |
| | Yes | 202 | 35.3 |
| | No | 370 | 64.7 |
| Attitude in talking about death | | | |
| | Feeling uncomfortable | 158 | 27.6 |
| | Trying to avoid | 74 | 12.9 |
| | Quite open | 34 | 59.4 |

Note: DCSS, Death Coping Self-Efficacy Scale; ICU, intensive care unit.

**Table 2. Mean death-coping self-efficacy scale scores of the nurses (N = 572).**

| Variable | Total mean score | Mean parameter score |
|---|---|---|
| Hospice care | 47.43±5.81 | 3.95 ±0.48 |
| Coping with grief | 27.14±4.50 | 3.39± 0.56 |
| Preparation for death | 28.01±5.58 | 3.11±0.62 |
| Total DCSS score | 102.58±12.07 | 3.48±0.42 |

**Table 3. Descriptive statistics of DCSS and its subscales, namely hospice care, coping with grief, preparation for death and other parameters.**

| Item No | Subdomain | Item | Mean (SD) |
|---------|-----------|------|-----------|
| 4 | Hospice Care | Listen to the family of a dying patient | 4.15(0.623) |
| **11** | **Hospice Care** | **Allow a patient to communicate fully** | **4.30(0.593)** |
| 16 | Hospice Care | Visit a dying friend | 4.16(0.653) |
| 25 | Hospice Care | Tolerate spiritual and religious differences | 4.25(0.571) |
| 3 | Hospice Care | Listen to the concerns of a dying patient | 4.21(0.575) |
| 5 | Hospice Care | Identify the concerns of a dying patient and his/her family | 3.91(0.665) |
| 17 | Hospice Care | Provide emotional support for the patient's family | 3.96(0.643) |
| 27 | Hospice Care | Care for me if I am experiencing stress in caring for a dying patient | 3.90(0.636) |
| 28 | Hospice Care | Be with a person at the time of death | 3.79(0.717) |
| 10 | Hospice Care | Ask to know if someone close to you has a terminal illness | 3.83(0.803) |
| 20 | Hospice Care | Attend a funeral or wake where the casket is open | 3.59(0.939) |
| 1 | Hospice Care | Be sensitive to the needs of the patient | 3.48(0.716) |
| 21 | Coping with Grief | Understand bereavement and grief | 4.01(0.724) |
| 6 | Coping with Grief | Handle the illness of your child | 3.89(0.712) |
| 26 | Coping with Grief | Cope with the death of a pet | 3.56(0.821) |
| 7 | Coping with Grief | Handle knowing that a family member has a fatal condition | 3.29(0.801) |
| 24 | Coping with Grief | Cope with the death of a friend the same age as you | 3.21(0.805) |
| 15 | Coping with Grief | Cope with the death of your father | 2.64(1.024) |
| **22** | **Coping with Grief** | **Cope with the death of your child** | **2.41(1.012)** |
| 23 | Coping with Grief | Handle the death of your spouse | 2.47(1.022) |
| 13 | Coping with Grief | Cope with the death of your mother | 2.53(1.034) |
| 2 | Preparation for Death | Buy life insurance | 3.62(0.900) |
| 9 | Preparation for Death | Listen to a news report of multiple deaths | 3.78(0.753) |
| 18 | Preparation for Death | Write a living will | 3.47(0.858) |
| 14 | Preparation for Death | Ask to know if you have a terminal illness | 3.83(0.803) |
| 8 | Preparation for Death | Prepare your will | 3.51(0.844) |
| 19 | Preparation for Death | Plan your funeral service | 3.03(0.869) |
| 29 | Preparation for Death | Prepay your funeral | 2.92(0.818) |
| 12 | Preparation for Death | Purchase your cemetery plot | 2.98(0.869) |

Note: DCSS, Death-coping Self-efficacy Scale.

## Factors affecting coping with grief

The scores of nurses on the coping with grief subscale were taken as the dependent variable, and factors with significance in general data were taken as independent variables. Multiple stepwise regression analysis was conducted with $\alpha = 0.05$ and $\alpha = 0.10$ in the entry and exit models, respectively. After the final entry into the equation, the factors were respectively analysed for age, attendance in palliative care education courses within the previous year and attitudes about death (Table 6).

**Table 4. Regression analysis of the DCS of nurses.**

| Independent variables | B | SE | β | t | P |
|-----------------------|---|----|----|---|---|
| Constant | 68.565 | 4.130 | - | 16.600 | <0.001 |
| Attending palliative care education courses within one year | -2.980 | 0.901 | -0.136 | -3.749 | <0.001 |
| Personal bereavement experience | 2.380 | 0.921 | -0.118 | -3.237 | <0.001 |
| Attitude in talking about death | 3.182 | 0.504 | 0.145 | 3.966 | <0.001 |

**Table 5. Regression analysis of the DCS of nurses.**

| Independent variables | B | SE | β | t | P |
|---|---|---|---|---|---|
| Constant | 4.346 | 0.156 | - | 27.826 | <0.001 |
| educational background | -0.153 | 0.065 | -0.095 | -2.334 | 0.020 |
| attended palliative care education courses within one year | -0.170 | 0.041 | -0.171 | -4.196 | <0.001 |
| Attitude in talking about death | 0.068 | 0.023 | 0.122 | 2.998 | 0.003 |

**Table 6. Regression analysis of the DCS of nurses.**

| Independent variables | B | SE | β | t | P |
|---|---|---|---|---|---|
| Constant | 2.674 | 0.151 | - | 17.720 | <0.001 |
| Age | 0.259 | 0.102 | 0.103 | 2.535 | 0.012 |
| attended palliative care education courses within one year | -0.129 | 0.052 | -0.101 | -2.483 | 0.013 |
| Attitude in talking about death | 0.147 | 0.029 | 0.208 | 5.118 | <0.001 |

**Table 7. Regression analysis of the DCS of nurses.**

| Independent variables | B | SE | β | t | P |
|---|---|---|---|---|---|
| Constant | 3.249 | 0.071 | - | 45.709 | <0.001 |
| length of service | 0.113 | 0.053 | 0.089 | 2.131 | 0.033 |

## Factors affecting preparation for death

The scores of nurses in the preparation for death subscale were taken as the dependent variable, and factors with significance in general data were taken as independent variables. Multiple stepwise regression analysis was conducted with α = 0.05 and α = 0.10 in the entry and exit models, respectively. After the final entry into the equation, the factors were analysed for the score of the length of service (Table 7).

## Discussion

The results of this study revealed that the score of each parameter in the DCSS was 3.48 ± 0.42, with the mean score of 3 as an intermediate criterion. This finding indicates that the overall DCSS score of the nurses was at a moderate level. With a higher average score for hospice care and the lowest score for coping with grief. Nurses, who have attended palliative care education courses within the previous year, have experienced accompanying the family members of the deceased, have an open attitude towards death, have higher self-efficacy in coping with death.

In this study, the score on the coping with grief subscale was the lowest and consistent with the results of a previous study [5], which indicated that their ability to cope with the death of their loved ones should improve. Most studies have supported that older people are more receptive to death than younger people, which was related to the clinical work experience of the study sample [2, 5, 20]. A study of nurses in palliative care units revealed that as nurses grew older and had more work experience, their self-efficacy in coping with hospice care, grief and death increases [19]. Kapoor et al. [21] found that some nurses believed that professionals must cover up the sadness and that the expression of sadness might be regarded as 'unprofessional' and 'weak'. Nurses want formal support; otherwise, they could not deal with their work emotionally and would rely on the informal networks of colleagues and friends outside the unit to talk about work. Bandura's self-care theory [22] suggested that nurses should clarify the

influence of self-loss or death experience on themselves, adjust their self-identity, find the meaning of events through the death experience and accept social support from others. Moreover, nurses' sense of life, psychological distress, job burnout and low negative emotions [23] suggest that clinical nurses, in addition to focusing on solving patients' problem, should form a group or end-of-life care facility for life and death issues, which should assist with emotional relief and signify their existence and value.

The main factors influencing the DCS of nurses were attendance in palliative care education courses within the previous year, experience of accompanying the family members of the deceased and attitude towards death.

In this study, age and clinical experience did not show independent significance after the adjustment for the basic attributes by multiple linear regression analysis. Possible reasons were as follows: our study participants were mostly young and junior nurses and the hospital investigated provided hospice joint nursing in 2019 and gave great importance to the palliative care training of nurses of various professional titles and levels. A study [24] also found that the nurses, working in intensive care units, with religious beliefs had better abilities of near-death management, death thinking and expression and life examination. However, this study found that religious affiliations had no effects on the DCS; the possible reason is that 94.2% of the participants in this study had no religious affiliations. Therefore, age, clinical work experience, religious beliefs and other factors had little influence on the DCS of nurses, which should be confirmed in studies with a large number of participants.

The nurses, who attended palliative care education courses within the last year, had higher DCSS scores, which was consistent with the results of Kim et al. [25] and Evenblij et al. [15]. This finding suggests that palliative care education courses could be an effective strategy to improve the DCS of nurses. Yang et al. [8] reported that the higher the number of end-of-life care courses the nurses received, the more they could understand the course and stages of death, recognise the emotions and needs of patients who are dying and positively think about death-related facts. Dehghani et al. [26] conducted a palliative care training programme in Iran, in which 40 nurses were randomly selected and trained four times. Each session lasted for 45 min. Through the questionnaire data analysis, they proved that the training significantly improved the nurses' sense of self-efficacy. White et al. [27] supported and trained nurses remotely using video conferencing, a network communication platform, and group discussions, for 6 months of 2-h palliative care teaching, and case discussion and suggested that nurses' knowledge and self-efficacy of palliative care were significantly improved after training. Hospitals should conduct practical education and training on topics where nurses have low knowledge levels, such as symptom management and spiritual care, to improve nursing knowledge and self-efficacy [28]. Education and training related to end-of-life care should be held regularly, especially regarding death counselling, stress management, self-affirmation and other related courses [26, 29] to help nurses understand and deal with their feelings of death when facing patients with terminal illness and prevent negative emotions such as fear and anxiety [30].

The experience of accompanying the family members of the deceased is one of the factors influencing the DCS scores of nurses, which was consistent with the results of Cheung et al. [4] and Ay et al. [31]. This might be because nurses could project their rich experience of end-of-life care to the patients, thereby showing their caring behaviour with empathy. Some studies [20] have also shown that contact with the palliative care team and experience in accompanying the family members of the deceased can provide more care to dying patients in the aspect of their psychology and spirituality, discussion of non-resuscitation (DNR) and palliative care [12].

The results of this study indicate that attitude towards death is an important factor that affects the DCS of nurses. Among them, the nurses scored the highest for open acceptance and the lowest for fear of death. The possible reason was that nurses felt frustrated in diagnosing and treating the dying patients and felt that it was not easy to establish a relationship with the family members of the deceased, thereby hoping that they could not face the death of patients [12]. A relevant study [32] revealed that nurses' fears included the fear of losing close people, sadness upon seeing their relatives and friends became sick, uncertainty about the time of death of their patients and subsequent treatment of death events (fear of body care, funeral planning, etc.). The open acceptance of nurses' attitudes towards death was positively correlated with their attitude towards dying patients and work engagement [2]. The fear of death and escape were negatively correlated with the meaning of nurses' life and tended to cause negative emotions [33]. In addition, clinical departments often faced the powerlessness of dying patients, and the inability to treat disease could result in emotional distress among nurses and thus job burnout feelings [2]. Burnout feelings among nurses could directly affect the quality of care and professional academic performance [34] and resulted in low DCS and demand. Moreover, the positive attitude towards end-of-life care was positively correlated with the ability to solve end-of-life-related problems [35].

According to the findings from our study, factors correlated with the level of educational background, attendance in palliative care education courses within the previous year and attitude towards death. Among the values of each subscale, hospice care recorded the highest average score, with 3.95 ± 0.48 points. Effective care can promote health, and the art of nursing is mainly expressed through caring [36]. A study showed that nursing care is positively correlated with satisfaction; the higher the frequency of nurses with caring behaviour, the higher would be the satisfaction of patients receiving nursing care [13]. Ling et al. [37] suggested that the priority of palliative care was to provide patients and their families comfort, satisfaction of needs, healthy environment, emotional support and protection of privacy and respect. Moreover, nursing itself is a representative group of the helping industry; therefore, a therapeutic environment with a caring atmosphere should be created. Through the characteristics of respect, focus, and care, the patients and their families can be assisted to resolve negative emotional reactions and respect the needs of the patients [38].

We found that the length of service (years), namely, ≤10 and >10, were the only significant independent variables with a significant association (p<0.05) with the preparation for death. The possible cause was that Chinese cultural background tends to avoid conversations about death. Therefore, nurses were reluctant to face this issue and were less prepared about their death, such as wills or plans for their future funerals [37]. Moreover, studies have shown that nurses rarely talk about death and lack knowledge and skills about death communication [39, 40]. China and Taiwan amended the palliative care medical regulations in 2013 and implemented the Patient Autonomy Rights Law in 2019, making the concept of palliative care, not performing non-cardiopulmonary resuscitation and gradually paying attention to the concept of pre-existing medical decisions. Patients can pre-sign and choose not to receive life-saving medical treatment or remove ineffective ones. Patients can think about not accepting invalid medical treatments at the end of their lives as early as possible, and in clinical practice, the medical staff can discuss with the patients and their families about shortening the suffering of patients at the end of their lives.

Based on social learning theory, Bandura developed self-efficacy, and proposed self-coordinated motivation, thinking strategies and behavioral paths to reflect appropriate behaviors and accomplish goals [22]. Medical staff's self-assessment of self-strength and performance shows that when they have low self-efficacy, they are more likely to compromise when confronted with adversity and obstacles, and often feel anxious, worried and frustrated. People with high

self-efficacy are willing to face difficulties and setbacks. And find out the appropriate method [41]. Studies [42, 43] have shown that high self-efficacy can improve the quality of care and ultimately improve individual and organizational performance. Scholars [44] found that during the coronavirus disease-2019 (COVID-19) pandemic, an assessment by nurses found that the ward nurses experienced higher than normal levels of despair, social isolation, and physical symptoms due to grief. Japanese scholars found that only 27% of nurses performed palliative care family psychological support care [45]. Dahlin [46] also pointed out that nurses in oncology wards lacked effective nursing methods to solve the psychological problems and negative emotions of end-stage patients, as well as lack of coping methods to alleviate the suffering of patients. In the three dimensions of death coping self-efficacy, the scores from high to low are Hospice Care, Preparation for Death, and Coping with Grief, which indicates that in clinical practice nurses pay more attention to the physical symptoms of patients, while the psychological needs of patients and caregivers often overlooked. This suggests that nurses' psychological and spiritual nursing self-efficacy needs to be strengthened urgently. Therefore, this paper can fully tap the potential of nurses, improve the efficiency and quality of nursing work, and promote the development of human health by understanding the self-efficacy level of nurses' death coping and discussing its influencing factors, and adopting targeted and effective methods.

## Limitations

Among the limitations of this study, a self-assessment questionnaire was adopted, which could be affected by personal cognition and social and cultural expectations or limitations. Future studies should conduct long-term in-depth interviews for qualitative analysis or retrospective research, and individual differences in mortality should be considered to achieve impeccable results. International studies are mostly cross-sectional studies that cannot dynamically observe the changes in death-coping ability and determine the causal relationship between death-coping ability and influencing factors. More scholars are expected to pay attention to the death-coping ability of nurses.

In terms of participants, some of the healthcare workers who currently belong to the selected setting were less numerous, for example: male nurses, nurses older than 40 years, non-religious nurses, etc. They may also have issues with Death-coping self-efficacy. However, in China, the absolute majority of hospital nurses are women, and only 2% of non-religious nurses are male. In addition, older nurses are generally not clinical, so in this study, we aimed to include only those only those clinical females, young and non-religious nurses, which suggests that this study cannot infer the current status of DCS for all clinical nurses.

International studies are mostly cross-sectional studies, which cannot dynamically observe the changes of death coping ability and determine the causal relationship between death coping ability and influencing factors. It is expected that more scholars will pay attention to the death coping ability of nurses.

## Conclusions

Studies [12, 37] show that when nurses perform palliative care, they often focus on physical care, but ignore the psychological and spiritual care of patients and their families. Nursing researchers should attach importance to nurses' palliative care self-efficacy, pay attention to psychological support, and improve nurses' psychological quality; set up relevant palliative care courses to promote nursing students to have a positive understanding of death; improve nurses' social support and strengthen palliative care psychology Nursing awareness, so as to improve nurses' palliative care self-efficacy and palliative care quality, thereby improving the comfort of dying patients, and promoting the development of palliative care.

## Supporting information

**S1 Data. The dataset from which the results of the study were produced (SPSS file).**
(ZIP)

## Acknowledgments

The researchers would like to express their gratitude to all nurses who contributed to this study and to Ms. Xiang for her support in data collection.

## Author Contributions

**Data curation:** Xi Lin, Weilan Xiang.

**Investigation:** Xiaoqin Li, Qin Liu.

**Methodology:** Xi Lin, Yongqi Bai.

**Resources:** Weilan Xiang.

**Software:** Xi Lin, Xiaoqin Li, Yongqi Bai.

**Supervision:** Yongqi Bai, Qin Liu.

**Writing – original draft:** Xi Lin.

**Writing – review & editing:** Yongqi Bai, Qin Liu.

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
