## [Decision Letter · Decision Letter 0]

5 Jul 2022

PONE-D-22-10472Death-coping Self-efficacy and its influencing factors among Chinese Nurses: A cross-sectional studyPLOS ONE

Dear Dr. Liu,

Thank you for submitting your manuscript to PLOS ONE. After careful consideration, we feel that it has merit but does not fully meet PLOS ONE’s publication criteria as it currently stands. Therefore, we invite you to submit a revised version of the manuscript that addresses the points raised during the review process.

We look forward to receiving your revised manuscript.

Kind regards,

Mohanad Mousa Taha Odeh

Academic Editor

PLOS ONE

Journal Requirements:

Reviewers' comments:

Reviewer's Responses to Questions

**Comments to the Author**

1. Is the manuscript technically sound, and do the data support the conclusions?

Reviewer #1: Yes

Reviewer #2: Yes

2. Has the statistical analysis been performed appropriately and rigorously? 

Reviewer #1: I Don't Know

Reviewer #2: Yes

3. Have the authors made all data underlying the findings in their manuscript fully available?

Reviewer #1: Yes

Reviewer #2: Yes

4. Is the manuscript presented in an intelligible fashion and written in standard English?

Reviewer #1: Yes

Reviewer #2: Yes

5. Review Comments to the Author

Reviewer #1: As a reviewer, I believe the study met all of the journal's publication criteria. However, I do have some concerns and suggestions for the manuscript.

1. In general it is a clear, well-written manuscript, The researcher's goal was to analyze the current state of clinical nurses' self-efficacy to cope with patients death and discuss it is influencing factors, but the sample was limited to almost females, young and non-religious nurses, Nurses with hospice care education, family accompanying experience, and an open attitude toward death have higher self-efficacy in coping with death, according to the findings.

2. in my opinion generalizing the findings to "clinical nurses" is an overestimation.

I believe that these limitations might be changed as a strength point that distinguishes those results from the current literature.

3. No distinction was made between palliative and hospice care, which are two distinct terms.

4. Excluding questionnaires that contained >10% missing data can lead to some bias, additionally, it was not mentioned how it was dealt with questionnaires that contained less than 10% of missing data.

5. What does "anonymous method" in line 190 mean?

6. it is stated that Parameter 25 has the highest mean (4.30) in line 250, whereas in Table 3, parameter 11 has a mean of (4.30) and parameter 25 has a mean of (4.30).

What impact did this have on the calculations and the outcomes?

7. On line 173, it is stated that there are eight questions regarding death preparation, and above that, it is stated that there are 29 parameters, but parameter 14 is missing in the death preparation section!

What impact did this have on the calculations and the outcomes?

8. The mean of parameter 25 is higher than the average, according to lines 383-385. Why is this parameter specifically mentioned in the paragraph while other parameters mean are higher than the average value but aren't mentioned in the paragraph?

9. finally, please Supply professional title for each author.

Regards

Reviewer #2: Thank you for allowing me the opportunity to review this journal submission and to provide feedback. This article to investigate death-coping Self-efficacy and its influencing factors among Chinese Nurses: A cross-sectional study.

This journal submission is about a topic that requires additional investigation. Nonetheless, some issues need to be addressed as follow:

Feedback:

Introduction:

The authors need to restructure the literature review to better make the case for why the data they are gathering is useful. Though there may be relatively little data collected about participants in the studied area. also few studies were addressed in this section

Methods:

1. Were there any exclusion criteria?

2. Sample and setting - more information on the context of sample collection would be helpful so that the reader can determine if the study applies to their population of interest. This is especially true since not all readers will be familiar with the demographic characteristics or hospital settings of China.

3. Please say more about the privacy and confidentiality procedures that were used

4. Study measures – please say more about what modifications were made to instruments

Discussion

1. The discussion's structure should be updated. Instead than repeating specific data and results, the authors should focus on larger themes. This will be easier to accomplish if the material they reference in the discussion is reviewed first in the literature review section.

2. many statements in the discussion section should be moved to a separate section named study implication

6. PLOS authors have the option to publish the peer review history of their article (what does this mean?). If published, this will include your full peer review and any attached files.

Reviewer #1: **Yes: **Ahmad Alazzam

Reviewer #2: **Yes: **Manar M AlAzzam

---

## [Author Response · Author response to Decision Letter 0]

21 Aug 2022

Dear Reviewers,

On behalf of my co-authors, I thank you very much for giving us an opportunity to revise our manuscript entitled “Death-coping Self-efficacy and its influencing factors among Chinese Nurses: A cross-sectional study” (PONE-D-22-10472R1). We appreciate you very much for your positive and constructive comments and suggestions. Those comments are all valuable and very helpful for revising and improving our paper, as well as the important guiding significance to our researches. 

We have carefully considered all issues mentioned in the reviewers' comments, outlined every change made point by point. We have made correction which we hope meet with approval. Revised portion are marked in red in the paper. The main corrections in the paper and the responds to the reviewer’s comments are as flowing:

Responds to the academic editor comments:

1.Please ensure that your manuscript meets PLOS ONE's style requirements, including those for file naming.

Response: We appreciate the reviewer very much for his positive comments and suggestions on our manuscript. We have revised the manuscript to conform to PLOS ONE's style requirements, including file naming.

Response: We have added financial disclosures at the end of the article and cover letter. ( line 517-518, pages 23)

3. In your Data Availability statement, you have not specified where the minimal data set underlying the results described in your manuscript can be found.

Response: We have expressed the data as Supporting information at the end of the article and uploaded it to Supporting information.

Response: We have removed ethical statements written outside of methods.

Reviewers' comments:

Response #1:

1. In general it is a clear, well-written manuscript, The researcher's goal was to analyze the current state of clinical nurses' self-efficacy to cope with patients death and discuss it is influencing factors, but the sample was limited to almost females, young and non-religious nurses, Nurses with hospice care education, family accompanying experience, and an open attitude toward death have higher self-efficacy in coping with death, according to the findings.

Response: We appreciate the reviewer very much for his positive comments and suggestions on our manuscript. In terms of participants, the selected setting had fewer healthcare workers, such as: male nurses, nurses older than 40, non-religious nurses, etc. They may also have issues with Death-coping self-efficacy. However, in China, the absolute majority of hospital nurses are women, and only 2% of non-religious nurses are male. In addition, older nurses are generally not clinical, so in this study, we aimed to include only those clinical females, young and non-religious nurses. This limit has been added to Limitations. ( line 453-457, pages 21)

2. In my opinion generalizing the findings to "clinical nurses" is an overestimation.

I believe that these limitations might be changed as a strength point that distinguishes those results from the current literature.

Response: Thank you very much for this helpful suggestion. We have changed "clinical nurses" to “nurses” and added a description at Limitations. ( line 453-457, pages 21)

3. No distinction was made between palliative and hospice care, which are two distinct terms.

Response: Thank you very much for this helpful suggestion. We have made changes in the article.

4. Excluding questionnaires that contained >10% missing data can lead to some bias, additionally, it was not mentioned how it was dealt with questionnaires that contained less than 10% of missing data.

Response: We are very sorry for the expression is not very clear in this paper. We excluded data that contained >10% missing data because the questionnaire lacked too many participants' attitudes, and the questionnaire had little reference value. After the questionnaire was collected on the spot, the completeness and logic of the questionnaire were tested. For <10% of the missing answers, they should be corrected in time. An explanation has been added in the Quality control. We excluded >10% of the missing data because the questionnaire was missing too many participants. The attitude of the respondents is not serious, and the reference value of the questionnaire is not large. After the questionnaire is collected on the spot, the completeness and logic of the questionnaire will be tested. If the answers are missing <10%, they should be corrected in time. An explanation has been added in Quality control. (line 208-210, pages 8)

5.What does "anonymous method" in line 190 mean?

Response: We are very sorry for the expression is not very clear in this paper，and we have removed the expression we have removed the expression.

6. it is stated that Parameter 25 has the highest mean (4.30) in line 250, whereas in Table 3, parameter 11 has a mean of (4.30) and parameter 25 has a mean of (4.30).

What impact did this have on the calculations and the outcomes?

Response: We are very sorry, due to the author's negligence, the average value of parameter 11 in line 252 should be (4.30), and the average value of parameter 25 is the highest (4.25), which has been modified at the table. This has no effect on the rest of the article, just in the article A typo in the form. (line 252-253, pages 12)

7. On line 173, it is stated that there are eight questions regarding death preparation, and above that, it is stated that there are 29 parameters, but parameter 14 is missing in the death preparation section!

Response: We are very sorry for the expression is not very clear in this paper，and we have supplemented parameter 14 in the table. There is parameter 14 in the original data, but we forgot to write it on the table. (Pages 13)

8. The mean of parameter 25 is higher than the average, according to lines 383-385. Why is this parameter specifically mentioned in the paragraph while other parameters mean are higher than the average value but aren't mentioned in the paragraph?

Response: We appreciate this suggestion, and We have removed this sentence from the article.

9. Finally, please Supply professional title for each author.

Response: Under your advice, we have added at Author contributions.

Reviewer #2:

1.he authors need to restructure the literature review to better make the case for why the data they are gathering is useful. Though there may be relatively little data collected about participants in the studied area. also few studies were addressed in this section

Response: We appreciate the reviewer very much for his positive comments and suggestions on our manuscript. We have restructured literature reviews to better illustrate why the data they collected are useful. (line 252-253, pages 12)

2.Methods:. Were there any exclusion criteria?

Response: We appreciate this suggestion. Exclusion criteria: ①Clinical nurses on shifts, advanced studies or going out; ②Nurses on sick leave, maternity leave or vacation. (line 144-145, pages 6)

3. Sample and setting - more information on the context of sample collection would be helpful so that the reader can determine if the study applies to their population of interest. This is especially true since not all readers will be familiar with the demographic characteristics or hospital settings of China.

Response: Under your advice, we have added inclusion criteria. (line 141-143, pages 6)

4. Please say more about the privacy and confidentiality procedures that were used.

Response: We are very sorry for the expression is not very clear in this paper. Confidentiality: After completing the questionnaire, the questionnaire and the subject's consent shall be sealed in an envelope and returned by the researcher himself. In order to protect the privacy of the case, in addition to filling in the questionnaire anonymously, the data collected will be numbered. The coding table and the questionnaire data will be stored separately, and the coding table will be properly kept, which is not accessible to others except researchers and professors. This study is subject to the personal Data Protection Act and the relevant laws and regulations, and will be carefully protected by the privacy of researchers. The research materials will only be used for the purpose of writing papers and publishing academic journals. They will not be linked to other institutions and will not be provided to the public. The completed questionnaires will be destroyed by shredder two years after the publication of the papers, and the destruction process will be witnessed by a third party other than this study.

5. Study measures – please say more about what modifications were made to instruments.

Response: We are very sorry for the expression is not very clear in this paper. he original scale includes 44 items: ① daily life and work (13 items); ② coping with the death of others (9 items); ③ death preparation behavior (8 items); ④ caring for terminally ill patients and family members (10 items); ⑤Near-death and post-death physical care (4 items). The revised scale includes three subscales: "hospice" (12 items), "grief coping" (9 items), and "death preparation" (8 items).

Discussion

6. The discussion's structure should be updated. Instead than repeating specific data and results, the authors should focus on larger themes. This will be easier to accomplish if the material they reference in the discussion is reviewed first in the literature review section.

Response: This is a very valuable recommendation, we didn’t think about it before. We have updated the discussion structure, with changes highlighted in red. (line 418-442, pages 20)

7. many statements in the discussion section should be moved to a separate section named study implication

Response: We appreciate this suggestion. Due to word limit, this part of the statement has been incorporated into the conclusion. (line 466-474, pages 20)

We tried our best to improve the manuscript and made some changes in the manuscript. These changes will not influence the content and framework of the paper. And here we did not list the changes but marked in red in revised paper. We appreciate for Reviewers’ warm work earnestly, and hope that the correction will meet with approval.

 Once again, thank you very much for your comments and suggestions.

Looking forward to hearing from you. 

Thank you and best regards.

Yours sincerely,

 Xi Lin

---

## [Editor Report · Decision Letter 1]

30 Aug 2022

Death-coping Self-efficacy and its influencing factors among Chinese Nurses: A cross-sectional study

PONE-D-22-10472R1

Dear Dr. Liu,

We’re pleased to inform you that your manuscript has been judged scientifically suitable for publication and will be formally accepted for publication once it meets all outstanding technical requirements.

Kind regards,

Mohanad Mousa Taha Odeh

Academic Editor

PLOS ONE

---

## [Editor Report · Acceptance letter]

2 Sep 2022

PONE-D-22-10472R1 

Death-coping Self-efficacy and its influencing factors among Chinese Nurses: A cross-sectional study 

Dear Dr. Liu:

I'm pleased to inform you that your manuscript has been deemed suitable for publication in PLOS ONE. Congratulations! Your manuscript is now with our production department. 

Kind regards, 

on behalf of

Dr. Mohanad Mousa Taha Odeh 

Academic Editor

PLOS ONE